# The Utilization Value of Condensate Water from the Drying Process of *Lonicera japonica* via Metabolomics Analysis

**DOI:** 10.3390/metabo15090569

**Published:** 2025-08-25

**Authors:** Da Li, Jiaqi Zhang, Yining Sun, Chongchong Chai, Fengzhong Wang, Bei Fan, Long Li, Shuqi Gao, Hui Wang, Chunmei Yang, Jing Sun

**Affiliations:** 1School of Biology and Medicine, Beijing City University, Beijing 100094, China; lida.lnzyydx@163.com (D.L.); 13021119842@163.com (J.Z.); m13315885616@163.com (Y.S.); wanghui2015@bcu.edu.cn (H.W.); 2Institute of Food Science and Technology, Chinese Academy of Agricultural Sciences, Key Laboratory of Agro-Products Processing, Ministry of Agriculture and Rural Affairs, Beijing 100193, China; cccbucm@163.com (C.C.); wangfengzhong@caas.cn (F.W.); fanbei517@163.com (B.F.); llzgnydx@163.com (L.L.); 3National Center of Technology Innovation for Comprehensive Utilization of Saline-Alkali Land, Dongying 257345, China; 17260602096@163.com

**Keywords:** *Lonicerae japonicae flos*, widely targeted metabolome analysis, volatile metabolomics, differential metabolites, by-products

## Abstract

**Background:** *Lonicerae japonicae flos* (LJF), a traditional food and medicine with a history spanning thousands of years, undergoes drying as a critical processing step in modern applications after regular processing. While the by-products of this process are typically discarded as waste, the potential value of LJF condensate water (JYHC) remains largely unexplored. To address this gap and investigate its potential utilization, this study conducted widely targeted metabolome and volatile metabolomics profiling analyses of ‘JYHC’. **Methods:** This study analyzed the differential metabolites of ‘JYHC’ and dried *Lonicerae japonicae flos* (JYHG) based on widely targeted metabolomics using UPLC-MS/MS. Additionally, the metabolic differences between fresh *Lonicerae japonicae flos* (JYHX) and ‘JYHC’ based on GC-MS volatile metabolomics were comprehensively analyzed. **Results:** A total of 1651 secondary metabolites and 909 volatile metabolites were identified in this study. Among these, flavonoids and terpenoids were the predominant secondary metabolites, while esters and terpenoids dominated the volatile fraction. Further comparison of the ‘JYHC’ and ‘JYHG’ groups revealed that 58 differential metabolites with potential biological activities were significantly up-regulated, with the types being terpenoids, phenolic acids, and alkaloids, which included nootkatone, mandelic acid, sochlorogenic acid B, allantoin, etc. Notably, a total of 186 novel compounds were detected in ‘JYHC’ that had not been previously reported in LJF such as isoborneol, hinokitiol, agarospirol, 5-hydroxymethylfurfural, α-cadinol, etc. **Conclusions:** This study’s findings highlight the metabolic diversity of ‘JYHC’, offering new theoretical insights into the study of LJF and its by-products. Moreover, this research provides valuable evidence supporting the potential utilization of drying by-products from LJF processing, paving the way for further exploration of their pharmaceutical and industrial applications.

## 1. Introduction

*Lonicerae japonicae flos* (LJF), which originates from the dried buds of *Lonicera japonica* Thunb., is a member of the Caprifoliaceae family and is native to East Asia, including China, Korea, and Japan. This plant is sun-preferring and cold-hardy, and is now widely distributed across temperate to subtropical regions. *Lonicera japonica* Thunb. is a semi-evergreen climber; its branchlets, petioles, and peduncles are covered with yellowish-brown stiff hairs or soft pubescence. Its leaves are papery and predominantly ovate or lanceolate in shape. Its fragrant flowers are paired and form axillary structures at branchlet apices. LJF has been extensively utilized in Traditional Chinese Medicine (TCM) for thousands of years [1,2]. In China, it is used as healthy food, as well as in cosmetics and soft drinks, due to its pleasant fragrance and high safety profile [3]. Recent research has indicated that LJF is composed of essential oils, organic acids, flavonoids, and triterpenoids, which demonstrate various effects such as antimicrobial, anti-inflammatory, antiviral, and immune-regulatory properties. Its clinical applications are widely recognized [4,5,6,7].

Drying is considered a crucial step in the processing of LJF (Figure 1). Drying effectively extends the shelf life of LJF and improves the stability and concentration of its active ingredients. However, its aromatic components are often lost or chemically transformed during the drying process [8,9]. At the same time, condensate water (JYHC) is usually generated. It is still unclear whether this by-product contains bioactive compounds. It is typically discarded as waste, leading to significant environmental pollution and a substantial waste of resources.

In recent years, the valorization of food industry by-products and agricultural wastes has been the focus of research worldwide [10,11]. Efforts have been made to explore the potential applications of these by-products for energy and resource conservation [12,13,14,15]. Metabolomics has been identified as a qualitative and quantitative analysis of all metabolites from different organisms, samples, or tissues. It is widely used in biomedical, botanical, and food science research [16,17,18,19,20,21]. A widely targeted metabolomics analysis is an innovative method that merges the benefits of both non-targeted and targeted metabolomics. This technique has been extensively applied in the analysis of plant metabolites across different species, including *Zingiberis rhizoma* and its processed forms, as well as citronella before and after drying [22,23].

This research utilized UPLC-ESI-MS/MS-based broad-spectrum metabolome profiling to identify differential metabolites in ‘JYHC’ and ‘JYHG’ (dry *Lonicerae japonicae flos*). In addition, a comprehensive analysis of the metabolic differences between ‘JYHX’ (Fresh *Lonicerae japonicae flos*) and ‘JYHC’ was conducted using GC-MS-based volatile metabolomics. The differential metabolites were analyzed by utilizing multivariate methods for statistical analysis. The findings of this study can be used as a new theoretical reference for the study of LJF and its by-products. Meanwhile, it also provides valuable insights into the potential utilization of the by-products generated during the drying process of LJF.

## 2. Materials and Methods

### 2.1. Experimental Materials

In this study, fresh *Lonicerae japonicae flos* (JYHX) was collected on 27 June 2024 in Dakuang Town, Laiyang City, Yantai, Shandong Province, China, and samples were annotated as JYHX (JYH-202406) by Professor Fengzhong Wang of the Institute of Food Science and Technology, Chinese Academy of Agricultural Sciences, China. The drying process of LJF was carried out at Yantai Riga Energy Saving Technology Co., Ltd. (Yantai, China), which mainly produces drying equipment; JYHC is considered a by-product of this process. The drying process used the whole fresh LJF plant (Figure 1).

### 2.2. Sample Preparation for Metabolite Extraction

#### 2.2.1. Preparation of ‘JYHC’ Test Material

We removed the ‘JYHC’ sample from an −80 °C freezer, and thawed it on ice. We vortexed the sample for 30 s. Then, 18 mL of the sample was transferred into a designated centrifuge tube, which was frozen at −80 °C overnight in a refrigerator, before proceeding with vacuum freeze-drying. After freeze-drying, 300 μL of 70% methanol extract containing an internal standard (2-chlorophenylalanine, ≥98% purity, J&K Scientific, Shanghai, China) was added at a concentration ratio of 60 times. The mixture was vortexed for 15 min, and then subjected to ultrasound in an ice water bath (KQ5200E, Beijing, China) for 10 min. The solution was centrifuged at 12,000 r/min and 4 °C (5424R, Eppendorf, Shanghai, China) for 3 min. The supernatant was collected, filtered through a microporous membrane (0.22 μm), and stored in an injection bottle.

#### 2.2.2. Preparation of ‘JYHG’ Test Material

The ‘JYHG’ sample was vacuum freeze-dried and subsequently ground into a fine powder. An amount of 50 mg of the powdered sample was weighed and added to 1200 μL of 70% methanol extract containing the internal standard. The mixture was vortexed for 15 min, and then left in a refrigerator at 20 °C for 30 min. The solution was centrifuged at 12,000 r/min and 4 °C for 3 min. The supernatant was collected, filtered through a microporous membrane (0.22 μm), and stored in an injection bottle.

#### 2.2.3. Preparation of ‘JYHX’ Test Material

The collected ‘JYHX’ was pulverized into a fine powder using liquid nitrogen. A total of 500 mg of the powder was added to a 20 mL headspace vial containing a saturated NaCl solution (Agilent, Waltham, MA, USA) for headspace solid-phase microextraction (HS-SPME).

### 2.3. Secondary Metabolites Analysis by UPLC-ESI-MS/MS

An ultra-performance liquid chromatography–tandem mass spectrometry (UPLC-MS/MS) system equipped with a triple quadrupole-linear ion trap (QTRAP) mass analyzer was employed for the analysis of secondary metabolites. Chromatographic separation was achieved using an Agilent SB-C_18_ column (1.8 μm, 2.1 mm × 100 mm) at 40 °C with a flow rate of 0.35 mL/min and an injection volume of 2 μL. The mobile phases consisted of 0.1% formic acid in water (A) and 0.1% formic acid in acetonitrile (B), and the gradient program was set as follows: linear decrease from 95% A to 5% A within 0–9 min and maintained for 1 min; followed by recovery to the initial ratio (95% A) within 1.1 min and equilibration for 2.9 min. Mass detection was performed using electrospray ionization (ESI) operated in both positive and negative modes.

Mass spectrometry was performed using an electrospray ionization (ESI) source with the following parameter settings: ion source temperature of 500 °C, spray voltage of +5500 V/−4500 V (positive/negative ion mode), nebulizing gas (GSI), auxiliary gas (GSII) and curtain gas (CUR) pressures were set to 50, 60 and 25 psi, respectively. Collision-induced dissociation (CAD) was operated in high-sensitivity mode. Multi-reaction monitoring (MRM) was employed for the detection of target metabolites. The dissociation potential (DP) and collision energy (CE) were individually optimized for each transition. All MRM transitions were dynamically scheduled according to the expected retention time of each analyte to maximize detection sensitivity.

### 2.4. Volatile Analysis by GC-MS

Volatile organic compounds (VOCs) were analyzed using an Agilent 8890 gas chromatography system coupled to a 7000D mass spectrometry (GC-MS, Agilent Technologies, Santa Clara, CA, USA). Sample injection was performed in splitless mode at an inlet temperature of 250 °C. After a 5 min solvent delay, the separation was carried out on a DB-5MS capillary column (30 m × 0.25 mm × 0.25 μm, 5% phenyl-polymethylsiloxane stationary phase) using helium as the carrier gas at a constant flow rate of 1.2 mL/min.

The chromatographic separation was carried out using the following temperature program: initial temperature held at 40 °C for 3.5 min, and then increased at 10 °C/min. The chromatographic separation was performed with a programmed temperature ramp: initial temperature of 40 °C for 3.5 min, ramp to 100 °C at 10 °C/min, ramp to 180 °C at 7 °C/min, and finally, ramp to 280 °C at 25 °C/min and held for 5 min. Mass spectrometric detection was conducted in electron bombardment ionization (EI) mode at 70 eV. The temperatures of the ion source, quadrupole, and transmission line were set at 230 °C, 150 °C, and 280 °C, respectively. Data acquisition was performed in selective ion monitoring (SIM) mode to enhance the detection accuracy of the target compounds.

### 2.5. Multivariate Statistical Analysis

To investigate the accumulation patterns of germplasm-specific metabolites, multivariate statistical analysis was applied to the metabolomic dataset. PCA and OPLS-DA were employed for metabolic profiling and pattern recognition. The distinct accumulation patterns of metabolites were visualized through heatmaps. All these analyses were performed in the Metware Cloud online platform. Differential metabolites were screened using the following criteria: FC ≥ 2 or ≤0.5, VIP ≥ 1, and *t*-test *p*-value ≤ 0.05 (MetaboAnalyst 5.0 platform).

### 2.6. KEGG Pathway Analysis

The KEGG pathway database was utilized to identify and show differential metabolites.

## 3. Results

### 3.1. Overview of the Metabolites in ‘JYHC’

This study integrated targeted and volatile metabolomics to comprehensively profile the secondary metabolites in ‘JYHC’. A total of 1651 metabolites were detected by comprehensive targeted metabolomics and further classified into eight categories (Figure 2A). Flavonoids represented the most abundant class, with a content that is 1.4 times that of terpenoids, 1.7–1.8 times that of alkaloids/phenolic acids, and 3.6 times that of lignans/coumarins. As secondary dominant components, terpenoids have an advantage of 1.2–1.3 times over alkaloids and phenolic acids. Collectively, flavonoids and terpenoids constituted the dominant secondary metabolites in ‘JYHC’, accounting for 44.88% of the total detected metabolites.

Volatile metabolomics profiling identified 909 metabolites, which were categorized into 14 categories (Figure 2B). It can be seen that the main volatile components in ‘JYHC’ are esters and terpenoids, collectively accounting for 37.29% of the total. Their relative abundances were approximately 1.7- and 1.6- fold higher than that of ketones, 1.8- and 1.7-fold greater than heterocyclic compounds, and 2.1- and 2.0-fold above those of alcohols, respectively. Compared to hydrocarbons, esters and terpenoids were 3.0 and 2.9 times more abundant, whereas trace components such as halogenated hydrocarbons and nitrogenous compounds exhibited substantially lower levels—only 1/26 to 1/138 of the abundance of esters.

### 3.2. PCA and OPLS-DA Analysis

PCA analysis was performed on the secondary metabolites of ‘JYHG’ versus (vs.) ‘JYHC’ (Figure 3A) and the volatile metabolites of ‘JYHX’ vs. ‘JYHC’ (Figure 3B), respectively. The ‘JYHG’ vs. ‘JYHC’ and ‘JYHX’ vs. ‘JYHC’ groups exhibited distinct separation on both PC1 and PC2, indicating significant differences in metabolite composition between ‘JYHC’ and ‘JYHG’, as well as between ‘JYHC’ and ‘JYHX’.

Pairwise comparative OPLS-DA revealed significant metabolic differences between ‘JYHC’ and ‘JYHG’ (R^2^X = 0.956), as well as between ‘JYHC’ and ‘JYHX’ (R^2^X = 0.962). The R^2^Y and Q^2^ values of all models were close to 1 (Q^2^ > 0.9), demonstrating that the models had excellent stability and predictive ability. As shown in Figure 3C,D, the scoring plots clearly showed the apparent separation of ‘JYHC’ from the other two sample groups, supporting the robustness of the model and providing a solid basis for subsequent screening of differential metabolites using variable importance in projection (VIP) analysis.

### 3.3. Differential Metabolites Screening

Differential metabolites were selected based on a variable importance in projection (VIP) value ≥ 1 from the OPLS-DA model, combined with univariate statistical criteria including a fold change ≥ 2 or ≤0.5 and a *p*-value ≤ 0.05. The results of this multi-criteria screening are visually summarized in volcano plots, which highlight metabolites meeting all these thresholds. In total, 1355 metabolites showed significant differences between ‘JYHC’ and ‘JYHG’, with 159 being upregulated and 1196 downregulated (Figure 4A); 984 volatile metabolites were significantly altered ‘JYHC’ and ‘JYHX’ (Figure 4B, 586 upregulated, 398 downregulated). Focusing on of the top 20 differential metabolites (VIP ≥ 1) between ‘JYHC’ and ‘JYHG’ groups, allantoin was the only metabolite showing significant accumulation, while the other 19 secondary metabolites showed pronounced downregulation. These downregulated metabolites primarily consisted of phenolic acids, terpenoids, and alkaloids (Figure 4C). Among the top 20 differential volatile metabolites between the ‘JYHC’ and ‘JYHX’, six metabolites (octadecane, piperonyl isobutyrate, (E)-3-methylpenta-1,3-diene-5-ol, 3-(1-methyl-2-propenyl)-1,5-cyclooctadiene, β-eudesmol, and tetrahydro-2H-pyran-2-one) showed significant increases, while the remaining 14 compounds exhibited significant decreases (Figure 4D).

Based on the VIP values, the top 50 differential secondary metabolites and volatile differential metabolites were selected and visualized by a heat map (Figure 5A,B). Heat map analysis results indicated that there was a significant difference in the abundance of these compounds between the ‘JYHC’ vs. ‘JYHG’ and ‘JYHC’ vs. ‘JYHX’ groups.

The results demonstrated that the metabolite profiles of the two sample groups formed two main clusters of metabolites, indicating significant differences in the metabolites between the groups. Combined with a literature search, 58 potentially biologically active up-regulated differential metabolites were screened in the ‘JYHC’ and ‘JYHG’ group (Table 1). Terpenoids, phenolic acids, and alkaloids were identified as the predominant classes of upregulated metabolites in ‘JYHC’.

Furthermore, volatile metabolomic analysis revealed the presence of 186 different components in ‘JYHC’ that were absent in ‘JYHX’ and have not been previously reported in LJF (Table 2). These metabolites may be produced under the specific conditions or processing of ‘JYHC’ and could have potential functional significance. The newly identified volatile components are mainly terpenoids, esters, and alcohols. Among them, terpenoids account for the highest proportion (25.27%). The 10 volatile metabolites with putative bioactivities were selected for further comparison analysis (Figure 6).

### 3.4. KEGG Pathway Analysis

To systematically investigate the biological implications of the observed metabolic changes, KEGG pathway enrichment analysis was performed to integrate the differential metabolites into functional pathways. In the ‘JYHC’ vs. ‘JYHG’ comparison, differential metabolites were annotated and enriched in 56 KEGG pathways. Among these, “flavonoid biosynthesis” and “flavone and flavonol biosynthesis” were significantly enriched (*p* < 0.05) (Figure 7A). In contrast, no pathways showed significant enrichment in the ‘JYHC’ vs. ‘JYHX’ comparison. Nevertheless, several pathways, including “biosynthesis of various plant secondary metabolites” and “biosynthesis of secondary metabolites”, exhibited relatively high enrichment factors, suggesting potential biological relevance.

## 4. Discussion

Thermal drying is a conventional step in the processing of LJF, which may lead to the loss of bioactive constituents through condensation. To evaluate whether the condensate ‘JYHC’ possesses potential utility, this study analyzed the secondary metabolites and volatile components in ‘JYHC’ using metabolomics approaches.

In this study, the UPLC-MS/MS-based widely targeted metabolome was employed to profile the metabolites of ‘JYHC’ and ‘JYHG’. A total of 1651 secondary metabolites were detected, including flavonoids, terpenoids, alkaloids, phenolic acids, lignans, coumarins, quinones, tannins, etc. Flavonoids and terpenoids were identified as the predominant secondary metabolites in ‘JYHC’ (Figure 2A). Flavonoids and terpenoids are widely involved in biological defence and have important pharmacological roles in human health, including anti-inflammatory, antimicrobial, and antioxidant properties [24,25,26,27,28,29,30].

A comprehensive comparative analysis and identification of metabolic differences between ‘JYHC’ and ‘JYHX’ were performed using GC-MS. A total of 909 metabolites, including esters, terpenoids, ketones, heterocyclic compounds, alcohols, hydrocarbons, etc. (Figure 2B). Esters and terpenoids were established as the dominant volatile metabolites in ‘JYHC’. Terpenes are widely found in plant essential oils and have anthelmintic, antibacterial, and anti-inflammatory activities [31,32]. Esters are closely related to the plant aromas and are widely used in the food, cosmetic, and flavor industries [33,34]. The composition and content of volatile metabolites are significantly affected by drying methods. Sun drying is a traditional method of drying LJF. Research has found that, while sun-dried *Lonicerae Flos* exhibits a greater variety of volatile compounds, its overall content is lower and it primarily contains hydrocarbons [35]. This outcome may be attributed to prolonged exposure to sunlight and air, which increases susceptibility to oxidative and photolytic degradation. Currently, convective drying methods are the most widely used in the industrial production of LJF, including programmed temperature oven drying, heat-pump drying, and hot-air drying [36]. Research indicates that *Lonicerae flos* dried via programmed temperature oven drying contains higher levels of esters and terpenes compared to other high-temperature drying methods [35]. Heat-pump drying is an advanced drying technology characterized by low temperature, mild conditions, and stable operation. This method is particularly suitable for preserving temperature-sensitive volatile components, such as short-chain fatty acids and terpenes [37].

To elucidate the functional significance of the altered metabolome, this study focused on the upregulated metabolites. Based on extensive literature and database mining, 58 potentially biologically active upregulated differential metabolites were screened in the ‘JYHC’ and ‘JYHG’ group (Table 1). Terpenoids, phenolic acids, and alkaloids are the main upregulated differential metabolites in ‘JYHC’. Terpenoids include nootkatone, catalpol, and isosteviol, etc. Nootkatone is an aromatic sesquiterpenoid with insecticidal and anti-inflammatory activities [38,39,40]. It is often used as a natural insecticide [41]. Isosteviol has been served as a cardiomyocyte protector [42,43,44]. Catalpol exhibits cardiovascular protective, neuroprotective, and hepatoprotective effects that were linked to NF-κB/NLRP3, PI3K/Akt, VEGF-A/KDR, and Jak-Stat pathways [45,46,47,48,49]. Among them, phenolic acid components include mandelic acid, vanillic acid, aloesone, isochlorogenic acid B, etc. Studies have shown that they possess biological activities such as anti-inflammatory and antibacterial effects [50,51,52]. Mandelic acid is widely used as a vital component in antiseptics and cosmetics due to its excellent antibacterial and anti-inflammatory activities [53]. Isochlorogenic acid B is recognized as a key active component in LJF. It exhibits significant blood sugar-lowering effects [54]. The alkaloids mainly include allantoin, betaine, solavetivone, aurantiamide, and stachydrine. Allantoin has activities such as antibacterial effects and skin damage repair, making it used in the development of cosmetic products [55,56,57,58]. Evidence has shown that betaine has beneficial actions in several human diseases and has anti-inflammatory functions in many diseases [59,60,61]. Stachydrine is a compound with anti-inflammatory activity, reducing inflammation through the P65, JAK2/STAT3, and NF-κB signaling pathways [62,63,64].

Additionally, 186 previously unreported compounds were identified in ‘JYHC’ that have not been documented in LJF, many of which demonstrate insecticidal, antimicrobial, and anti-inflammatory activities (Table 2). Terpenoids accounted for the highest percentage of newly identified volatile components. Terpenoids generally have some antibacterial and anti-inflammatory activity; for example, isoborneol and hinokitiol have good antimicrobial activity and are used in the application and development of natural antimicrobial materials [65,66,67]. Agarospirol, α-cadinol have been proven to be the main anti-inflammatory active ingredient in natural plant essential oils [68,69,70]. Curcumin has significant anti-inflammatory activity and can treat osteoarthritis by inhibiting NF-κB and MAPK pathways [71]. Moreover, 5-Hydroxymethylfurfural is a common reaction product during heat processing with good anti-inflammatory activity [72,73]. Esculetin is a natural dihydroxy coumarin, which has been shown to prevent cell growth in a variety of cancers [74,75,76]. Palmitoleic acid has been shown to regulate the gut microbiota and promote intestinal homeostasis [77]. However, it should be noted that this study is limited by the lack of absolute quantification for some differential metabolites. Future work will focus on validating these results through the construction of a more comprehensive standard compound library.

## 5. Conclusions

This research utilized UPLC-ESI-MS/MS-based widely targeted metabolomics method and GC-MS-based volatile metabolomics to comprehensively profile the metabolites in ‘JYHC’. The results shown that ‘JYHC’ is abundant in diverse secondary and volatile metabolites. Flavonoids and terpenoids were identified as the predominant secondary metabolites, whereas esters and terpenoids dominated the volatile fraction. Comparative metabolomic analysis revealed 58 upregulated differential secondary metabolites, primarily consisting of terpenoids, phenolic acids, and alkaloids. Additionally, 186 previously unreported compounds were identified in LJF, including isoborneol, hi-nokitiol, Agarospirol, and α-cadinol, etc. These metabolites are suggested to possess potential biological activities, such as anti-inflammatory and antimicrobial effects.

In the future, ‘JYHC’ shows potential for application as a functional additive with antibacterial and anti-inflammatory properties in cosmetics, or as a plant-derived biopesticide in green agriculture. The comprehensive utilization of ‘JYHC’ could enhance the economic value of LJF, while simultaneously reducing resource waste and wastewater pollution associated with its traditional processing. This approach aligns closely with the principles of green and sustainable development.

## Figures and Tables

**Figure 1 metabolites-15-00569-f001:**
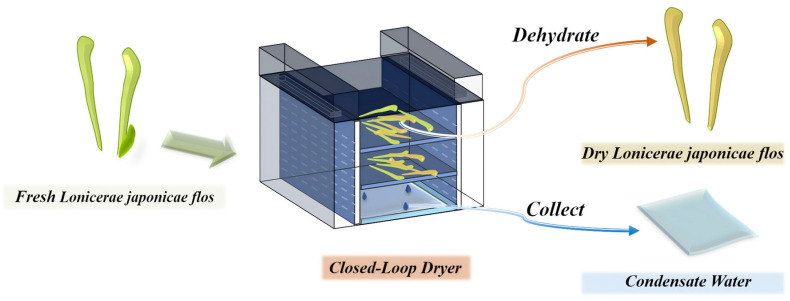
*Lonicerae japonicae flos* drying process flowchart.

**Figure 2 metabolites-15-00569-f002:**
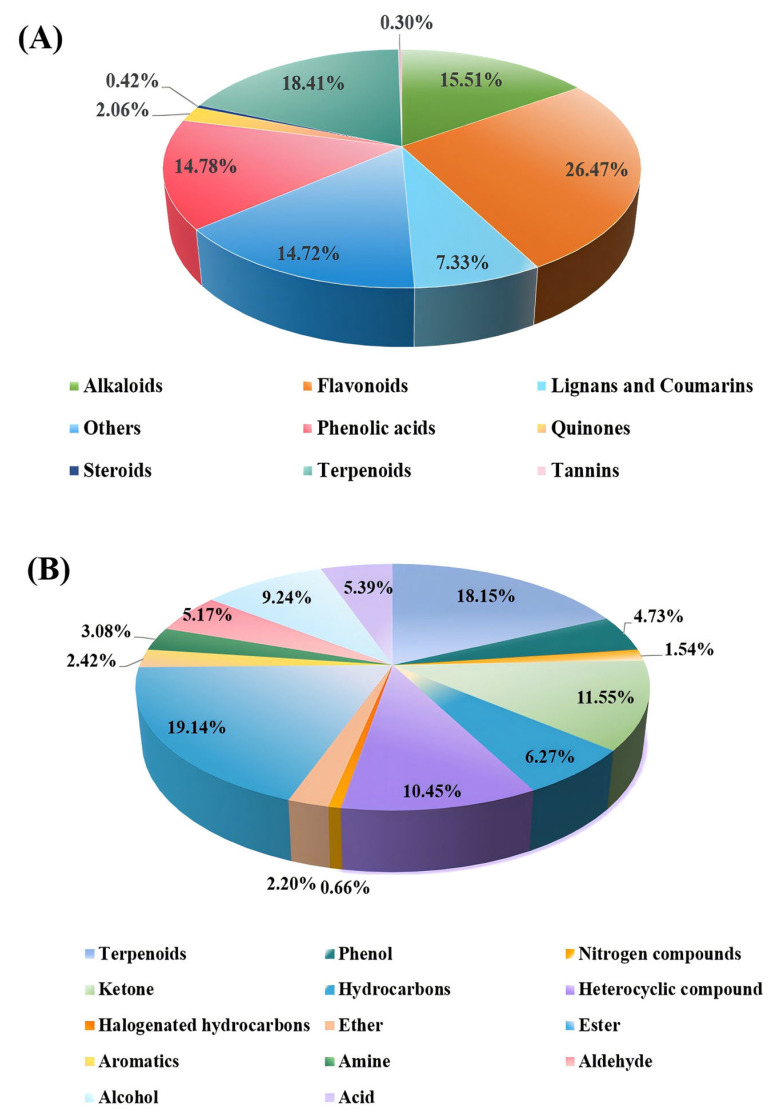
Classification of the metabolic profiles in ‘JYHC’. (**A**): the major secondary metabolites in ‘JYHC’ (**B**): volatile metabolomics in ‘JYHC’.

**Figure 3 metabolites-15-00569-f003:**
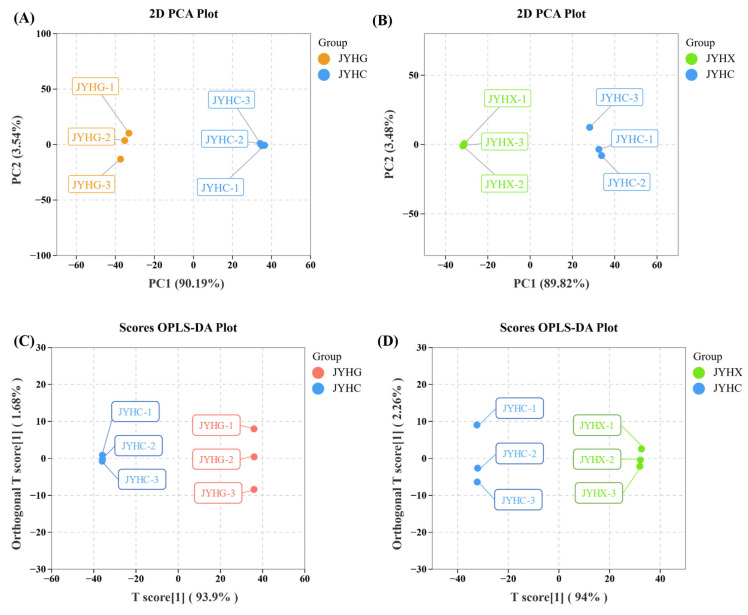
Principal component analysis (PCA) and score plots generated from orthogonal partial least squares discriminant analysis (OPLS-DA) of ‘JYHC’ compared with ‘JYHX’/‘JYHG’ (**A**): PCA score plot in ‘JYHC’ vs. ‘JYHG’ (**B**): PCA score plot in ‘JYHC’ vs. ‘JYHX’ (**C**): OPLS-DA score plot in ‘JYHC’ vs. ‘JYHG’ (**D**): OPLS-DA score plot in ‘JYHC’ vs. ‘JYHX’.

**Figure 4 metabolites-15-00569-f004:**
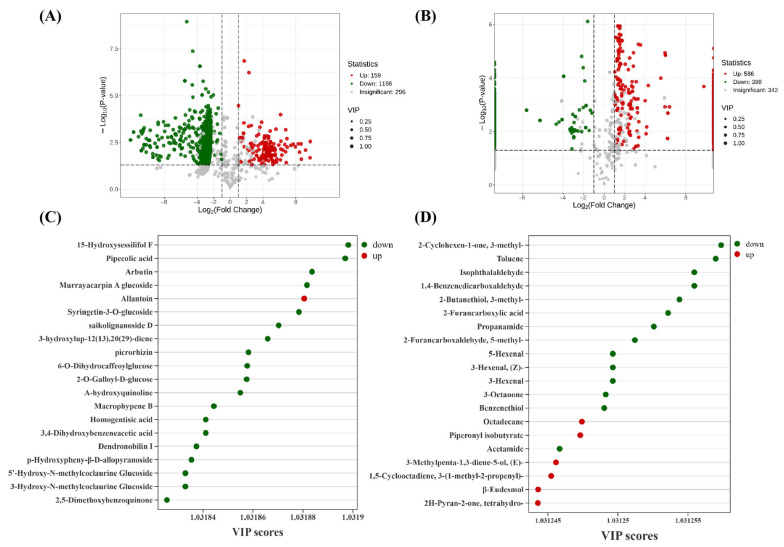
Differential metabolite results of ‘JYHC’ compared with ‘JYHX’/‘JYHG’ (**A**): Volcano plots showing the levels of the differential metabolites in ‘JYHC’ vs. ‘JYHG’ (**B**): Volcano plots showing the levels of the differential metabolites in ‘JYHC’ vs. ‘JYHX’ (**C**): Variable importance in the project (VIP) plot of the top 20 differential secondary metabolites identified by OPLS-DA in ‘JYHC’ vs. ‘JYHG’ (**D**): Variable importance in the project (VIP) plot of the top 20 differential volatile secondary metabolites identified by OPLS-DA in ‘JYHC’ vs. ‘JYHX’.

**Figure 5 metabolites-15-00569-f005:**
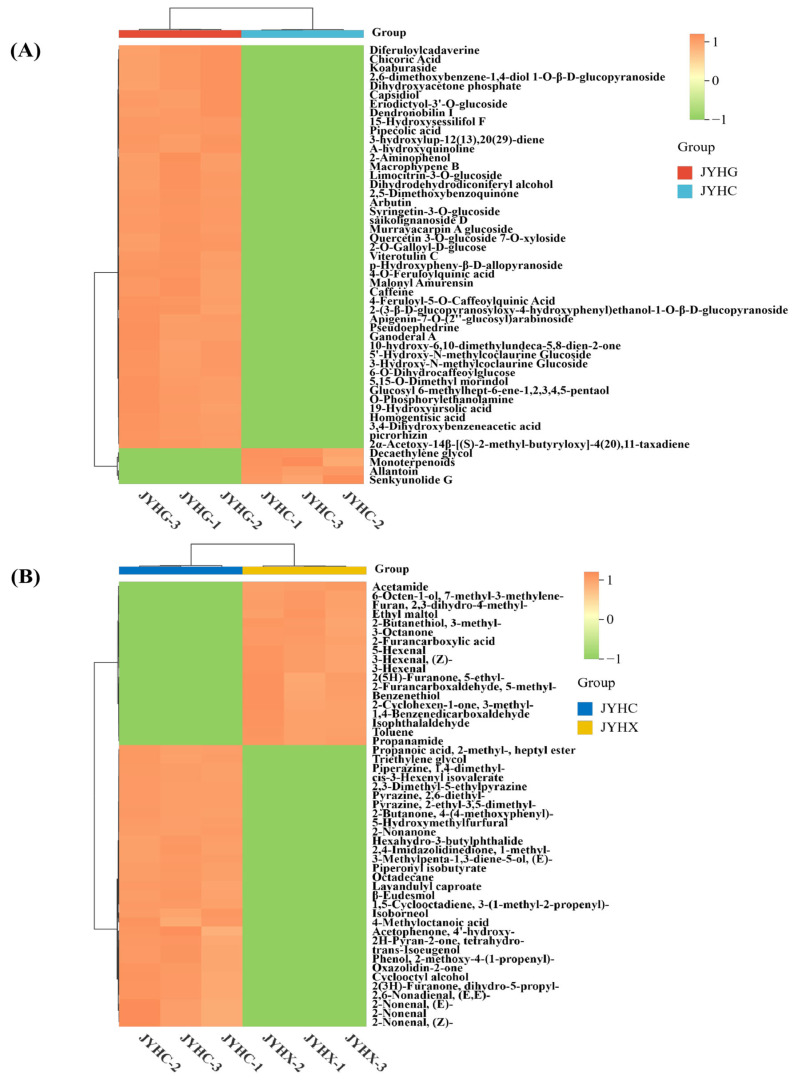
A heat map of top 50 differential metabolites. (**A**): Heatmap is based on relative abundance of differential metabolites in ‘JYHC’ vs. ‘JYHG’ (**B**): Heatmap is based on relative abundance of differential metabolites in ‘JYHC’ vs. ‘JYHX’.

**Figure 6 metabolites-15-00569-f006:**
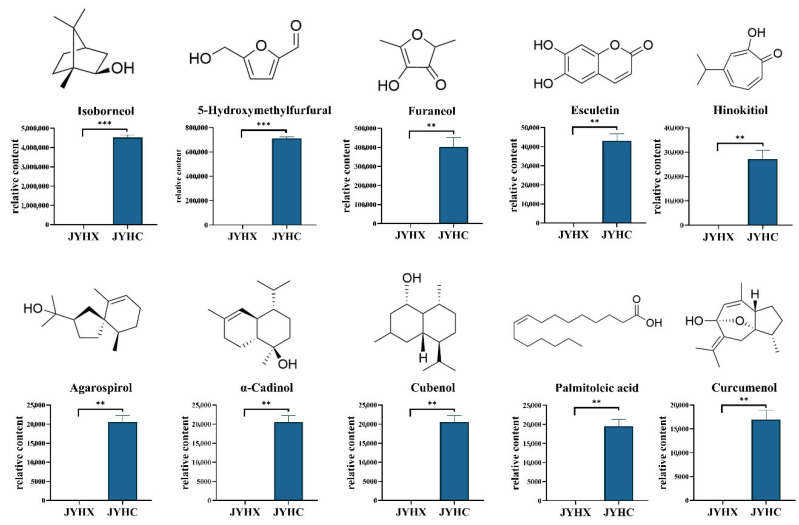
Chemical structures and quantities of 10 significant differential volatile metabolites in ‘JYHX’ and ‘JYHC’. The value bars with asterisks denote statistically significant differences: ** *p* < 0.01; *** *p* < 0.001.

**Figure 7 metabolites-15-00569-f007:**
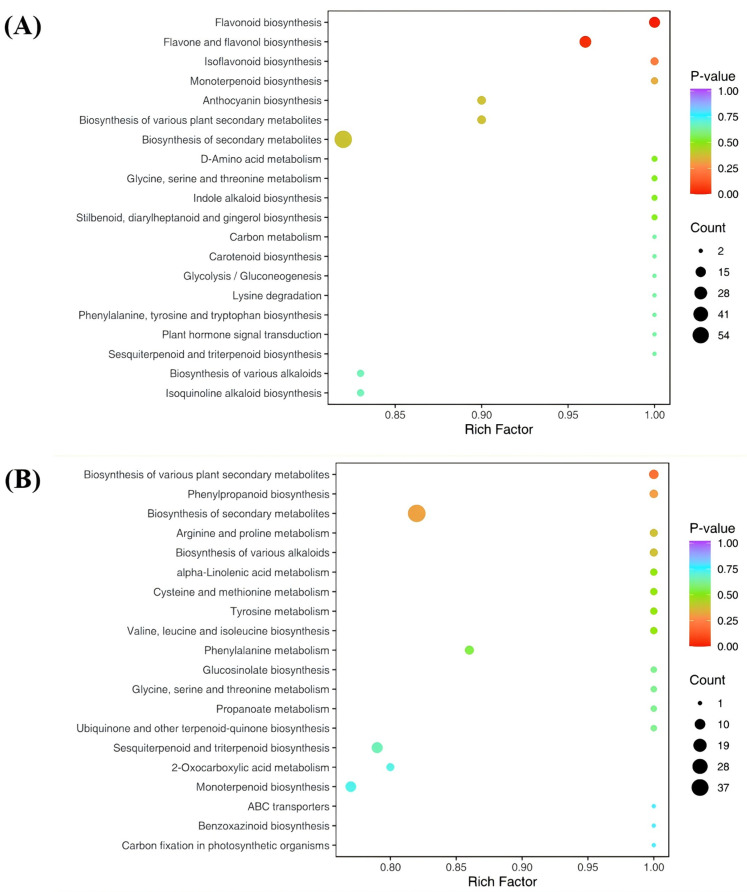
Kyoto Encyclopedia of Genes and Genomes (KEGG) annotations and enrichment results. (**A**): ‘JYHC’ vs. ‘JYHG’ (**B**): ‘‘JYHC’’ vs. ‘JYHX’.

**Table 1 metabolites-15-00569-t001:** The information on differential metabolites in ‘JYHC’ and ‘JYHG’ groups.

No	Class	Compounds	VIP	*p*-Value	FC
1	Terpenoids	11-acetylamarolide	1.0314	0.0029	30.3923
2		Epikatonic acid	1.0310	0.0043	19.8401
3		Isosteviol	1.0309	0.0037	14.8007
4		Verbenalin; Cornin	1.0301	0.0037	5.5482
5		Ilicic acid	1.0301	0.0074	16.6428
6		Catalpol	1.0300	0.0131	45.3682
7		Curcumenone	1.0284	0.0160	23.8513
8		7-Oxodehydroabietic acid	1.0283	0.0037	318.2749
9		Monomelittoside	1.0283	0.0222	32.3884
10		Nootkatone	1.0267	0.0189	15.8299
11		26,27-Dihydroxylanosta-7,9(11),24-trien-3-one	1.0252	0.0265	13.4889
12		Pisiferic acid	1.0185	0.0561	17.3450
13		Oleuropeic acid	1.0114	0.0246	7.5915
14	Phenolic acids	4-Methylbenzoic acid	1.0315	0.0020	25.1366
15		Mandelic acid	1.0314	0.0043	60.8357
16		3-Methylsalicylic Acid	1.0312	0.0040	28.0947
17		Aloesone	1.0311	0.0071	71.9037
18		2-Methylbenzoic acid	1.0309	0.0062	41.9502
19		Phthalic acid	1.0295	0.0058	33.8591
20		Propyl 4-hydroxybenzoate	1.0295	0.0139	29.7368
21		2-(Formylamino)benzoic acid	1.0295	0.0003	14.3260
22		Piperonylic acid	1.0290	0.0141	23.0521
23		4-Hydroxybenzoic acid	1.0255	0.0062	6.0468
24		Methyl 4-hydroxybenzoate	1.0227	0.0048	10.3296
25		Vanillic acid	1.0219	0.0010	3.1129
26		3-Hydroxy-4-methoxybenzoic acid; Isovanillic Acid	1.0165	0.0068	4.3149
27		Dihydroferulic Acid	1.0132	0.0277	11.6718
28		Benzoic acid	1.0092	0.0083	4.8672
29		Isochlorogenic acid B	1.0003	0.0723	8.4708
30	Alkaloids	Allantoin	1.0319	0.0001	70.7447
31		2,4,6,6-Tetramethyl-3(6H)-pyridinone	1.0317	0.0028	872.5247
32		6,7-dimethoxy-2-methyl-3,4-dihydro-1h-isoquinoline	1.0316	0.0014	18.2081
33		Solavetivone	1.0310	0.0053	33.1317
34		2(3H)-Benzothiazolone	1.0308	0.0078	44.7512
35		Hordenine	1.0307	0.0060	22.9234
36		Aurantiamide acetate	1.0306	0.0055	18.0396
37		Betaine	1.0305	0.0030	15.7049
38		Candicine	1.0304	0.0247	411.4936
39		Aurantiamide	1.0303	0.0075	20.8872
40		Isatin	1.0301	0.0148	79.3473
41		Stachydrine	1.0291	0.0019	17.9690
42		Dopamine	1.0286	0.0018	2.8634
43		N-Caffeoylputrescine	1.0283	0.0147	16.4331
44		N-Methylserotonin	1.0272	0.0209	17.4776
45		Cinnamamide	1.0261	0.0365	44.7934
46		N-Isobutyl Decanamide	1.0124	0.0096	7.1458
47		N-Butylbenzenesulfonamide	1.0043	0.0101	20.8540
48		3-Hydroxypyridine	1.0009	0.0328	4.9551
49	Quinones	2,7-Dihydroxy-4-methoxyphenanthrene	1.0315	0.0016	18.1935
50		anthraquinone-2-aldehyde	1.0309	0.0046	18.8696
51		Hircinol(2,5-Dihydroxy-4-methoxy-9,10-dihydrophenanthrene)	1.0282	0.0175	19.3460
52		2-Methyl-1,3,6-trihydroxy-9,10-anthraquinone	1.0281	0.0174	23.5978
53		2,3-Dihydro-1,4-naphthoquinone	1.0042	0.0006	47.7627
54	Flavonoids	6-prenylnaringenin	1.0280	0.0377	114.6009
55		Homomangiferin	1.0274	0.0351	50.2422
56		4′-Hydroxy-2,4,6-trimethoxydihydrochalcone; Loureirin B	1.0274	0.0274	35.3058
57		Herbacetin	1.0264	0.0353	37.1602
58		Chrysin	1.0048	0.0314	44.3305

**Table 2 metabolites-15-00569-t002:** The information on differential metabolites in ‘JYHC’ and ‘JYHX’ groups.

No	Class	Compounds	VIP	*p*-Value	FC
1	Terpenoids	1-((1S,3aR,4R,7S,7aS)-4-Hydroxy-7-isopropyl-4-methyloctahydro-1H-inden-1-yl)ethanone	1.0307	0.0041	inf
2		Caryophyllenyl alcohol	1.0301	0.0128	inf
3		2,6-Dimethyl-2-trans-6-octadiene	1.0309	0.0049	inf
4		2,6-Octadiene, 2,6-dimethyl-	1.0309	0.0049	inf
5		Agarospirol	1.0311	0.0022	inf
6		5,9-Undecadien-2-one, 6,10-dimethyl-	1.0003	0.1224	inf
7		Naphthalene, 1,6-dimethyl-4-(1-methylethyl)-	1.0304	0.0063	inf
8		Hexadecane, 2,6,10,14-tetramethyl-	1.0312	0.0014	inf
9		1,6,10-Dodecatriene, 7,11-dimethyl-3-methylene-	1.0279	0.0597	inf
10		α-Cadinol	1.0311	0.0022	inf
11		τ-Cadinol	1.0311	0.0022	inf
12		cis-α-Bisabolene	1.0310	0.0013	inf
13		2,4,6-Cycloheptatrien-1-one, 2-hydroxy-4-(1-methylethyl)-	1.0308	0.0058	inf
14		1-Cyclohexene-1-carboxylic acid, 4-(1-methylethenyl)-	1.0312	0.0014	inf
15		Dill ether	1.0311	0.0007	inf
16		7-epi-α-Eudesmol	1.0312	0.0004	inf
17		τ-Muurolol	1.0312	0.0008	inf
18		Cubenol	1.0311	0.0022	inf
19		3-Buten-2-one, 4-(2,2,6-trimethyl-7-oxabicyclo[4.1.0]hept-1-yl)-	1.0311	0.0023	inf
20		Benzofuran, 4,5,6,7-tetrahydro-3,6-dimethyl-	1.0312	0.0004	inf
21		Cyclohexane, 1-ethenyl-1-methyl-2,4-bis(1-methylethenyl)-, [1S-(1.alpha.,2.beta.,4.beta.)]-	1.0303	0.0122	inf
22		Geranyl tiglate	1.0306	0.0045	inf
23		β-sesquiphellandrene	1.0304	0.0121	inf
24		(3S,3aR,3bR,4S,7R,7aR)-4-Isopropyl-3,7-dimethyloctahydro-1H-cyclopenta[1,3]cyclopropa[1,2]benzen-3-ol	1.0301	0.0206	inf
25		Cyclohexene, 4-[(1E)-1,5-dimethyl-1,4-hexadien-1-yl]-1-methyl-	1.0310	0.0013	inf
26		Citronellyl tiglate	1.0312	0.0015	inf
27		cis-β-Farnesene	1.0279	0.0597	inf
28		α-Farnesene	1.0310	0.0040	inf
29		(E)-β-Farnesene	1.0279	0.0597	inf
30		2-Cyclohexen-1-ol, 2-methyl-5-(1-methylethenyl)-, cis-	1.0312	0.0003	inf
31		Cyclohexanol, 5-methyl-2-(1-methylethyl)-, [1S-(1.alpha.,2.alpha.,5.beta.)]-	1.0311	0.0013	inf
32		4-Hexen-1-ol, 5-methyl-2-(1-methylethenyl)-, (R)-	1.0312	0.0004	inf
33		Isoborneol	1.0312	0.0003	inf
34		Bicyclo[2.2.1]heptane, 2-chloro-1,7,7-trimethyl-, (1R-endo)-	1.0281	0.0223	inf
35		Cyclohexanol, 3-ethenyl-3-methyl-2-(1-methylethenyl)-6-(1-methylethyl)-, [1R-(1.alpha.,2.alpha.,3.beta.,6.alpha.)]-	1.0304	0.0059	inf
36		2-Furanmethanol, tetrahydro-.alpha.,.alpha.,5-trimethyl-5-(4-methyl-3-cyclohexen-1-yl)-, [2S-[2.alpha.,5.beta.(R*)]]-	1.0301	0.0073	inf
37		Benzene, 1-(1,5-dimethyl-4-hexenyl)-4-methyl-	1.0312	0.0013	inf
38		2-Cyclohexen-1-ol, 3-methyl-6-(1-methylethyl)-	1.0182	0.0940	inf
39		Curcumenol	1.0309	0.0041	inf
40		β-Caryophyllene Alcohol	1.0301	0.0128	inf
41		Benzene, 1-methyl-4-(1,2,2-trimethylcyclopentyl)-, (R)-	1.0307	0.0062	inf
42		Salvial-4(14)-en-1-one	1.0307	0.0074	inf
43		(R,1E,5E,9E)-1,5,9-Trimethyl-12-(prop-1-en-2-yl)cyclotetradeca-1,5,9-triene	1.0301	0.0095	inf
44		1,7-Dimethyl-4-(propan-2-ylidene)tricyclo[4.4.0.02,7]decan-3-one	1.0311	0.0007	inf
45		(1R,5S)-1,8-Dimethyl-4-(propan-2-ylidene)spiro[4.5]dec-7-ene	1.0308	0.0029	inf
46		Geranyl isobutyrate	1.0300	0.0124	inf
47		Bicyclo[3.2.1]oct-2-ene, 3-methyl-4-methylene-	1.0307	0.0152	inf
48	Ester	Pentanoic acid, pentyl ester	1.0301	0.0108	inf
49		Hexanoic acid, 3-hexenyl ester	1.0301	0.0147	inf
50		2,6-Octadien-1-ol, 3,7-dimethyl-, propanoate, (Z)-	1.0041	0.1340	inf
51		Lauryl acetate	1.0029	0.1300	inf
52		2,4-Hexadienoic acid, ethyl ester, (2E,4E)-	1.0153	0.1108	inf
53		5-Azulenemethanol, 1,2,3,4,5,6,7,8-octahydro-.alpha.,.alpha.,3,8-tetramethyl-, acetate, [3S-(3.alpha.,5.alpha.,8.alpha.)]-	1.0309	0.0026	inf
54		Benzoic acid, 2-hydroxy-, phenylmethyl ester	1.0311	0.0009	inf
55		Bicyclo[2.2.1]heptan-2-ol, 1,7,7-trimethyl-, formate, endo-	1.0312	0.0013	inf
56		cis-3-Hexenyl isovalerate	1.0312	0.0001	inf
57		Isobutyl isovalerate	1.0306	0.0069	inf
58		Dodecanoic acid, ethyl ester	1.0094	0.2222	inf
59		(S)-4-(1-Acetoxyallyl)phenyl acetate	1.0311	0.0019	inf
60		Octanedioic acid, dimethyl ester	1.0311	0.0035	inf
61		2,6,10-Dodecatrienoic acid, 3,7,11-trimethyl-, methyl ester, (E,E)-	1.0310	0.0057	inf
62		Hexadecanoic acid, ethyl ester	1.0303	0.0096	inf
63		Hexyl tiglate	1.0304	0.0135	inf
64		Hexanoic acid, 3,7-dimethyl-2,6-octadienyl ester, (E)-	1.0312	0.0005	inf
65		Butanoic acid, 2-methyl-, 2-methylpropyl ester	1.0306	0.0069	inf
66		Propanoic acid, 2-methyl-, heptyl ester	1.0312	0.0001	inf
67		Undecanoic acid, ethyl ester	1.0311	0.0019	inf
68		Isobornyl formate	1.0312	0.0013	inf
69		2(3H)-Furanone, 5-hexyldihydro-	1.0310	0.0051	inf
70		6-Octen-1-ol, 3,7-dimethyl-, propanoate	1.0312	0.0032	inf
71		Butanoic acid, 3-hexenyl ester, (Z)-	1.0312	0.0007	inf
72		2(3H)-Furanone, dihydro-5-propyl-	1.0312	0.0007	inf
73		Propanoic acid, pentyl ester	1.0310	0.0030	inf
74		Pentadecanoic acid, 3-methylbutyl ester	1.0312	0.0007	inf
75		(Z)-Hex-3-enyl (E)-2-methylbut-2-enoate	1.0307	0.0043	inf
76		Butanedioic acid, diethyl ester	1.0312	0.0007	inf
77		Butanoic acid, 2-methyl-, pentyl ester	1.0301	0.0108	inf
78		.delta.-Nonalactone	1.0311	0.0027	inf
79		(Z)-3-Butylidene-4,5-dihydroisobenzofuran-1(3H)-one	1.0310	0.0021	inf
80		Propanoic acid, heptyl ester	1.0311	0.0018	inf
81		Neryl butyrate	1.0303	0.0126	inf
82		(3S,3aR)-3-Butyl-3a,4,5,6-tetrahydroisobenzofuran-1(3H)-one	1.0312	0.0014	inf
83		Butanoic acid, butyl ester	1.0306	0.0084	inf
84		Butanoic acid, 3-hexenyl ester, (E)-	1.0312	0.0007	inf
85	Alcohol	2-Nonen-1-ol, (Z)-	1.0312	0.0001	inf
86		3,7-Octadiene-2,6-diol, 2,6-dimethyl-	1.0296	0.0281	inf
87		n-Pentadecanol	1.0311	0.0044	inf
88		1-Undecanol	1.0055	0.2222	inf
89		1,2-Benzenediol, 3,4,5,6-tetrachloro-	1.0301	0.0070	inf
90		Benzenepropanol, 4-hydroxy-3-methoxy-	1.0312	0.0004	inf
91		Benzenemethanol, 4-hydroxy-	1.0312	0.0001	inf
92		1-Tetradecanol	1.0308	0.0046	inf
93		2-Tridecen-1-ol, (E)-	1.0223	0.1641	inf
94		2-Nonen-1-ol, (E)-	1.0312	0.0001	inf
95		Benzenemethanol, α-ethyl-	1.0311	0.0023	inf
96		5-Hexen-1-ol	1.0312	0.0008	inf
97		3-Nonanol	1.0312	0.0002	inf
98		1-Decanol	1.0306	0.0110	inf
99		1-Hexanol, 2-ethyl-	1.0306	0.0061	inf
100		2-Nonanol	1.0312	0.0003	inf
101		Bicyclo[3.1.1]hept-2-ene-2-ethanol, 6,6-dimethyl-	1.0201	0.0666	inf
102		1-Naphthalenemethanol	1.0266	0.0257	inf
103		n-Tridecan-1-ol	1.0288	0.0195	inf
104		4a(2H)-Naphthalenol, octahydro-4,8a-dimethyl-,(4.alpha.,4a.alpha.,8a.beta.)-	1.0065	0.1917	inf
105		2-Octanol	1.0292	0.0259	inf
106		2-Octanol, (S)-	1.0292	0.0259	inf
107		Cyclohexanol, 5-methyl-2-(1-methylethyl)-, (1.alpha.,2.beta.,5.beta.)-	1.0311	0.0017	inf
108		(E)-2,6-Dimethylocta-3,7-diene-2,6-diol	1.0296	0.0281	inf
109		Triethylene glycol	1.0312	0.0001	inf
110	Ketone	5,9-Undecadien-2-one, 6,10-dimethyl-, (E)-	1.0003	0.1224	inf
111		5,9-Undecadien-2-one, 6,10-dimethyl-, (Z)-	1.0003	0.1224	inf
112		Ethanone, 1-(2,4,6-trihydroxyphenyl)-	1.0204	0.1003	inf
113		2-Piperidinone	1.0161	0.2222	inf
114		9-Decen-2-one	1.0022	0.2222	inf
115		2-Nonanone	1.0312	0.0000	inf
116		2H-Pyran-2-one, tetrahydro-	1.0312	0.0002	inf
117		4-(N-Nitroso-N-methylamino)-1-(3-pyridyl)-1-butanone	1.0311	0.0023	inf
118		2-Cyclohexen-1-one, 4-(3-hydroxy-1-butenyl)-3,5,5-trimethyl-	1.0312	0.0009	inf
119		Tropinone	1.0312	0.0001	inf
120		7,9-Di-tert-butyl-1-oxaspiro(4,5)deca-6,9-diene-2,8-dione	1.0312	0.0006	inf
121		Acetophenone, 4′-hydroxy-	1.0312	0.0004	inf
122		2H-1-Benzopyran-2-one, 4-hydroxy-	1.0312	0.0005	inf
123		2,4-Imidazolidinedione, 1-methyl-	1.0312	0.0001	inf
124		3-Butylisobenzofuran-1(3H)-one	1.0312	0.0018	inf
125		2H-Pyran-2-one, 6-pentyl-	1.0312	0.0009	inf
126		Furaneol	1.0311	0.0045	inf
127		2-Undecanone	1.0307	0.0038	inf
128		2-Butanone, 4-(4-methoxyphenyl)-	1.0312	0.0001	inf
129		1-Pentanone, 1-(2-furanyl)-	1.0312	0.0005	inf
130		3,5,9-Undecatrien-2-one, 6,10-dimethyl-	1.0296	0.0136	inf
131		Ethanone, 2-hydroxy-1-phenyl-	1.0312	0.0004	inf
132		1-Propanone, 1-(4-methoxyphenyl)-	1.0308	0.0075	inf
133	Aldehyde	2-octenal	1.0306	0.0071	inf
134		(E)-Tetradec-2-enal	1.0308	0.0038	inf
135		2-Undecenal, E-	1.0291	0.0122	inf
136		2-Nonenal	1.0312	0.0007	inf
137		2-Nonenal, (Z)-	1.0312	0.0007	inf
138		5-Hydroxymethylfurfural	1.0312	0.0001	inf
139		Pentadecanal-	1.0311	0.0009	inf
140		3-p-Menthen-7-al	1.0312	0.0005	inf
141		2-Undecenal	1.0291	0.0122	inf
142		Isoneral	1.0312	0.0003	inf
143		2,6-Nonadienal, (E,Z)-	1.0312	0.0008	inf
144		2-Nonenal, (E)-	1.0312	0.0007	inf
145		Tridecanal	1.0310	0.0020	inf
146		cis-4-Decenal	1.0310	0.0044	inf
147		2-Octenal, (E)-	1.0306	0.0071	inf
148	Acid	2-Octenoic acid, (E)-	1.0308	0.0041	inf
149		2,3,4-Trihydroxybenzoic acid	1.0302	0.0084	inf
150		2-Octenoic acid	1.0308	0.0041	inf
151		Homovanillic acid	1.0312	0.0009	inf
152		Benzenepropanoic acid, 4-hydroxy-	1.0312	0.0010	inf
153		Propanoic acid, 3-(methylthio)-	1.0311	0.0014	inf
154		Palmitoleic acid	1.0309	0.0029	inf
155		4-Methyloctanoic acid	1.0312	0.0002	inf
156		Benzeneacetic acid, α-hydroxy-, (R)-	1.0312	0.0010	inf
157		3,7,11-Trimethyl-dodeca-2,6,10-trienoic acid	1.0308	0.0070	inf
158	Heterocyclic compound	Thiophene, 2-ethyl-	1.0312	0.0010	inf
159		1,3,5-Triazine-2,4,6-triamine	1.0310	0.0022	inf
160		Pyrazine, 2-ethyl-3,5-dimethyl-	1.0312	0.0001	inf
161		Pyrazine, 3-ethyl-2,5-dimethyl-	1.0312	0.0001	inf
162		5,6-Dihydro-5-methyluracil	1.0294	0.0221	inf
163		1H-Pyrazole	1.0310	0.0025	inf
164		2-n-Butyl furan	1.0302	0.0063	inf
165		Esculetin	1.0311	0.0023	inf
166		2,3-Dimethyl-5-ethylpyrazine	1.0312	0.0001	inf
167		2-Acetyl-3-methylpyrazine	1.0311	0.0022	inf
168		7-Oxabicyclo[4.1.0]heptane	1.0312	0.0014	inf
169		Quinoline, 2,4-dimethyl-	1.0312	0.0024	inf
170	Hydrocarbons	1-Heptadecene	1.0291	0.0207	inf
171		Dodecane	1.0305	0.0117	inf
172		Fucoserratene	1.0311	0.0023	inf
173		Hexadecane, 2-methyl-	1.0305	0.0048	inf
174		Heptadecane, 7-methyl-	1.0309	0.0021	inf
175		Heptadecane, 2-methyl-	1.0312	0.0011	inf
176	Phenol	2-Methoxy-5-methylphenol	1.0311	0.0015	inf
177		trans-Isoeugenol	1.0312	0.0006	inf
178		Phenol, 2-methoxy-4-(1-propenyl)-	1.0312	0.0006	inf
179		Creosol	1.0311	0.0015	inf
180		Phenol, 4-(3-hydroxy-1-propenyl)-2-methoxy-	1.0312	0.0021	inf
181		1-Naphthalenol	1.0309	0.0057	inf
182	Ether	Benzene, 1,3-dimethoxy-	1.0312	0.0005	inf
183		Benzene, 1,1′-[oxybis(methylene)]bis-	1.0312	0.0007	inf
184		1,3-Benzodioxole, 4-methoxy-6-(2-propenyl)-	1.0305	0.0200	inf
185		Asarone	1.0311	0.0019	inf
186		Benzene, 1-ethenyl-4-methoxy-	1.0312	0.0016	inf

Column FC, ‘inf’ means infinity.

## Data Availability

The data supporting the findings of this study are available from the corresponding author upon reasonable request.

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
