# Peer review of "The Utilization Value of Condensate Water from the Drying Process of Lonicera japonica via Metabolomics Analysis"

_metabolites, 2025, doi:10.3390/metabo15090569_

Round 1
Reviewer 1 Report
Comments and Suggestions for Authors
The manuscript is well-written, scientifically sound, and presents novel and valuable findings regarding the metabolomic profiling of Lonicera japonica drying by-products.
In addition to the overall positive assessment, I would like to offer the following specific suggestions for improvement:
Line 3 (Title): In the manuscript title, the scientific name of the plant species should follow the rules of binomial nomenclature: the genus name (Lonicera) is capitalized, while the species epithet (japonica) should be written in lowercase. Therefore, please revise “Lonicera Japonica” to “Lonicera japonica” in the title to ensure taxonomic accuracy and consistency with international scientific standards.
Line 37: I recommend avoiding the repetition of words from the manuscript title in the list of keywords. Including additional, non-redundant keywords can help improve the discoverability of your article in database searches.
Line 40: The Latin name of the herbal drug Lonicerae japonicae flos should be consistently italicized in its entirety. In the current manuscript, only the genus and species are italicized (Lonicerae japonicae), while flos is left in regular font. Since this is the official name of the herbal drug, including the plant part, all three words should be italicized to reflect standard pharmaceutical and pharmacognostic conventions.
Line 40-41: The botanical author citation "Thunb" should include a period, as it is the standardized abbreviation of Carl Peter Thunberg’s name. Therefore, the correct form is Lonicera japonica Thunb., in accordance with the International Code of Nomenclature for algae, fungi, and plants (ICN). Please revise accordingly.
Line 40-41: Please consider including the botanical classification of Lonicera japonica in the manuscript for clarity and completeness. Specifically, it would be helpful to mention that Lonicera japonica belongs to the family Caprifoliaceae. Adding this information, preferably in the introduction or in the first mention of the species, would provide readers with important taxonomic context.
Line 47: The term “honeysuckle” is a vernacular name that can refer to many species within the Lonicera genus. Since this study focuses specifically on Lonicera japonica, it would be more accurate and scientifically appropriate to refer to the species by its full Latin name, and to clarify that “honeysuckle” here specifically denotes Lonicera japonica. This will improve clarity for international readers and ensure terminological precision.
Line 76: Please consider providing additional details regarding the plant materials used in this study. Specifically, it would be beneficial to include the following information:
Exact dates of sample collection;
Voucher specimen numbers);
The name of the person who identified the plant specimens.
Including this information would enhance the transparency, traceability, and reproducibility of the research, and aligns with good practice standards for studies involving plant materials.
Author Response
Reviewer #1:
- Line 3 (Title): In the manuscript title, the scientific name of the plant species should follow the rules of binomial nomenclature: the genus name (Lonicera) is capitalized, while the species epithet (japonica) should be written in lowercase. Therefore, please revise “Lonicera Japonica” to “Lonicera japonica” in the title to ensure taxonomic accuracy and consistency with international scientific standards.
Response: We apologize for this oversight. We have changed “Lonicera Japonica” to “Lonicera japonica” in the revised manuscript line 3 which ensure taxonomic accuracy and consistency with international scientific standards.
- Line 37: I recommend avoiding the repetition of words from the manuscript title in the list of keywords. Including additional, non-redundant keywords can help improve the discoverability of your article in database searches.
Response: Thanks for your great comments and constructive suggestions. In the revised manuscript we have replaced the origin keywords “condensate water” with the “differential metabolites” and “by-products” to increase the discoverability of our article and avoiding the additional non-redundant keyword. (in Line 38)
- Line 40: The Latin name of the herbal drug “Lonicerae japonicae flos” should be consistently italicized in its entirety. In the current manuscript, only the genus and species are italicized (Lonicerae japonicae), while flos is left in regular font. Since this is the official name of the herbal drug, including the plant part, all three words should be italicized to reflect standard pharmaceutical and pharmacognostic conventions.
Response: We apologize for this problem, and the original file was rechecked. We have changed all instances of “Lonicerae japonicae flos” with “Lonicera japonica flos” throughout the revised manuscript to align with standard pharmacognosy and medicinal plant nomenclature conventions.
- Line 40-41: The botanical author citation "Thunb" should include a period, as it is the standardized abbreviation of Carl Peter Thunberg’s name. Therefore, the correct form is Lonicera japonica Thunb., in accordance with the International Code of Nomenclature for algae, fungi, and plants (ICN). Please revise accordingly.
Response: We apologize for this problem, and the original file was rechecked. We have revised “Lonicera japonica Thunb” to “Lonicera japonica Thunb.” (in lines 41-42) according to the International Code of Nomenclature for algae, fungi, and plants (ICN). We would like to express our sincere gratitude once again for your valuable suggestions.
- Line 40-41: Please consider including the botanical classification of Lonicera japonica in the manuscript for clarity and completeness. Specifically, it would be helpful to mention that Lonicera japonica belongs to the family Caprifoliaceae. Adding this information, preferably in the introduction or in the first mention of the species, would provide readers with important taxonomic context.
Response: Thanks for your great comments and constructive suggestions. We have changed “Lonicerae japonicae flos (LJF), originating from the dried buds of Lonicera japonica Thunb.” to “Lonicerae japonicae flos (LJF), originating from the dried buds of Lonicera japonica Thunb., which is in the Caprifoliaceae family.”, which shows Lonicera japonica belongs to the family Caprifoliaceae.
- Line 47: The term “honeysuckle” is a vernacular name that can refer to many species within the Lonicera genus. Since this study focuses specifically on Lonicera japonica, it would be more accurate and scientifically appropriate to refer to the species by its full Latin name, and to clarify that “honeysuckle” here specifically denote Lonicera japonica. This will improve clarity for international readers and ensure terminological precision.
Response: Thanks for your great comments and constructive suggestions. We have changed all the “honeysuckle” to “LJF” (Lonicerae japonicae flos), to clarity for international readers and ensure terminological precision. We would like to express our sincere gratitude once again for your valuable suggestions.
- Line 76: Please consider providing additional details regarding the plant materials used in this study. Specifically, it would be beneficial to include the following information: Exact dates of sample collection; Voucher specimen numbers; The name of the person who identified the plant specimens. Including this information would enhance the transparency, traceability, and reproducibility of the research, and aligns with good practice standards for studies involving plant materials.
Response: Thanks for your great comments and constructive suggestions. As for information of Lonicerae japonicae flos, we have already added “In this study, Fresh Lonicerae japonicae flos (JYHX) were collected on 27 June 2024 in Dakuang Town, Laiyang City, Yantai, Shandong Province. and were annotated as JYHX (JYH-202406) by Professor Fengzhong Wang of the Institute of Food Science and Technology, Chinese Academy of Agricultural Sciences, China” Please see line 84-87.
Reviewer 2 Report
Comments and Suggestions for Authors
The manuscript could not be recommended due to some major and minor errors in the manuscript text and the work methodology
- Some detailed results such as characteristic metabolites names should be presented in the abstract section.
- As a paper background, the author is encourage to prepare a brief concerned to
Lonicerae japonicae botanical and ecological attributes. - The author(s) could used a standard compound such as cappric acid to quantified all volatile compounds in the samples.
- It is essential to mention the quantitative value for each compound.
- The discussion is not sufficient. The author is suggested to provide more attention to comparison between different drying methods in volatile compounds profile.
- The multivariant statistics analyzed illustrations are not sufficient. more diagrams such as heat map and VIP scores should be provided in the manuscript.
Author Response
Reviewer #2:
- Some detailed results such as characteristic metabolites names should be presented in the abstract section.
Response: We sincerely appreciate the reviewer's valuable suggestion. In response, we have revised the abstract by: (1) adding the significantly upregulated characteristic metabolites “nootkatone, mandelic acid, sochlorogenic acid B, and allantoin” in the comparison between ‘JYHC’ and ‘JYHG’ groups (Lines 29-30); (2) supplementing representative bioactive compounds “including isoborneol, hinokitiol, agarospirol, 5-hydroxymethylfurfural, and α-cadinol” among the 186 novel compounds detected in ‘JYHC’ (Line 31-32). These modifications have further strengthened the completeness of our results presentation.
- As a paper background, the author is encourage to prepare a brief concerned to Lonicerae japonicae botanical and ecological attributes.
Response: We sincerely appreciate the reviewer's constructive comments. In revised manuscript, we have enriched the botanical and ecological description of Lonicerae japonicae flos (LJF) in the Introduction section (Line 41-47). The added content details are as follows:“Lonicerae japonicae flos (LJF), originating from the dried buds of Lonicera japonica Thunb., which is in the Caprifoliaceae family and native to East Asia, including China, Korea, and Japan. This plant is sun-preferring and cold-hardy, which is now widely distributed across temperate to subtropical regions. Lonicera japonica Thunb. is a semi-evergreen climber, its branchlets, petioles, and peduncles are covered with yellowish-brown stiff hairs or soft pubescence. Leaves are papery, predominantly ovate or lanceolate in shape. Fragrant flowers are paired and axillary at branchlet apices.” We would like to express our sincere gratitude once again for your valuable suggestions.
- The author(s) could used a standard compound such as cappric acid to quantified all volatile compounds in the samples.
Response: Thanks for your great comments and constructive suggestions. We fully acknowledge the importance of absolute quantification in metabolomics studies. However, complete quantification of all volatile compounds remains challenging due to the unavailability of some reference standards. We have added this deficiency to the Discussion section, and We will conduct further research and supplemental analyses on this aspect in future studies. We would like to express our sincere gratitude once again for your valuable suggestions.
- It is essential to mention the quantitative value for each compound.
Response: We sincerely appreciate the reviewer's constructive comment. In revised manuscript, we have systematically compiled the relative quantitative values for all secondary metabolites in ‘JYHC’ and ‘JYHG’ groups. (in Supporting Information 1); along with the corresponding data for volatile metabolites between ‘JYHC’ and ‘JYHX’ groups. (in Supporting Information 2).
- The discussion is not sufficient. The author is suggested to provide more attention to comparison between different drying methods in volatile compounds profile.
Response: We appreciate the reviewer's constructive comment. We have revised the discussion section accordingly and added comparative analysis regarding the effects of different drying methods on volatile compounds, as detailed in lines 281-292. “The composition and content of volatile metabolites are significantly affected by drying methods. Sun drying is a traditional method of drying LJF. Research has found that, while sun-dried Lonicerae Flos exhibits a greater variety of volatile compounds, its overall content is lower and it primarily contains hydrocarbons [35]. This outcome may be attributed to prolonged exposure to sunlight and air, which increases susceptibility to oxidative and photolytic degradation. Currently, convective drying methods are the most widely used in the industrial production of LJF, including programmed temperature oven drying, heat-pump drying, and hot-air drying [36]. Research indicates that Lonicerae flos dried via programmed temperature oven drying contains higher levels of esters and terpenes compared to other high-temperature drying methods [35]. Heat-pump drying is an advanced drying technology characterized by low temperature, mild conditions, and stable operation. This method is particularly suitable for preserving temperature-sensitive volatile components, such as short-chain fatty acids and terpenes [37].”
Reference:
- Wu, C.; Wang, F.; Liu, J.; Zou, Y.; Chen, X., A comparison of volatile fractions obtained from Lonicera macranthoides via different extraction processes: ultrasound, microwave, Soxhlet extraction, hydrodistillation, and cold maceration. Integr. Med. Res. 2015, 4, (3), 171-177.
- Li, S.-F.; Guo, X.-M.; Hao, X.-F.; Feng, S.-H.; Hu, Y.-J.; Yang, Y.-Q.; Wang, H.-F.; Yu, Y.-J., Untargeted metabolomics study of Lonicerae japonicae flos processed with different drying methods via GC-MS and UHPLC-HRMS in combination with chemometrics. Ind. Crops Prod. 2022, 186, 115179.
- Song, J.; Han, J.; Fu, L.; Shang, H.; Yang, L., Assessment of characteristics aroma of heat pump drying (HPD) jujube based on HS-SPME/GC–MS and e-nose. J. Food Compos. Anal. 2022, 110, 104402.
- The multivariant statistics analyzed illustrations are not sufficient. more diagrams such as heat map and VIP scores should be provided in the manuscript.
Response: We sincerely appreciate the reviewer's insightful suggestion. In revised manuscript, we have added two new figures (Figure 4C and Figure 4D) displaying VIP score plots of the 20 most significant volatile and secondary differential metabolites, respectively. Additionally, recognizing that the original heatmap was overly extensive and somewhat unclear in its presentation, we have replaced it with an improved version (Figure 5). The new heatmap separately illustrates the top 50 most significant volatile and secondary differential metabolites. These modifications significantly enhance the clarity and effectiveness of our data presentation.Please see word or manuscripts.

Reviewer 3 Report
Comments and Suggestions for Authors
The authors have carried out a serious study of undoubted practical significance, especially for pharmacology and medicine. The methods and results are described consistently, the text is well structured. Special thanks to the authors for the list of abbreviations used. However, I would like to make several recommendations to improve the comprehension of the text by readers.
- Lines 49 and 50. The word ‘however’ is repeated twice. In one of the sentences it can be removed without losing the meaning.
- Section 2.1 The authors indicated the province from which 'JYHX' were obtained. If possible, it would be very good if the authors also indicated where exactly the Lonicerae japonica plants grew. This is a rather important point, since the composition and content of bioactive substances in plants may vary depending on the growing conditions. Understanding these conditions will be useful for readers and other researchers. At the same time, if the authors do not have the necessary information, I do not insist on providing the above data.
- Line 86. The authors indicate that they used an internal standard. What substance/substances were used as an internal standard? This information should be provided with the company name.
- Results. Paragraph on lines 149-158. When describing the results, do not duplicate the data presented in the figures. Since the relative content of the substances studied by the authors has already been indicated in the diagrams, it is better to describe in the text how many times / by what % the content of one group of substances exceeds the content of another / other groups of substances.
- In the Discussion, mentions to figures and tables given in the text of the manuscript should be added in parentheses at the end of some sentences. For example, '(Fig. 2A)' should be added at the end of the sentence on lines 243-244. Such mentions to figures and tables will help readers understand where in the discussion the results of this study are present, and where the literature data are.
- Line 340 – 2015 is repeated twice.
- Line 376 – not all dots have been placed in the abbreviated title of the journal.
These comments do not reduce the scientific and practical value of the work and are aimed only at improving the quality of the manuscript and the perception of the text by readers.
Author Response
Reviewer #3:
- Lines 49 and 50. The word ‘however’ is repeated twice. In one of the sentences it can be removed without losing the meaning.
Response: Thanks for your great comments and constructive suggestions. We have removed the redundant 'however' in Line 58 and rephrased the sentence as: “However, the aromatic components are often lost or chemically transformed during the drying process [8, 9]. At the same time, condensate water (JYHC) is usually generated. It is still unclear whether this by-product contains bioactive compounds.” The modification has been highlighted in the revised manuscript. (in lines 56-58)
- Section 2.1 The authors indicated the province from which 'JYHX' were obtained. If possible, it would be very good if the authors also indicated where exactly the Lonicerae japonica plants grew. This is a rather important point, since the composition and content of bioactive substances in plants may vary depending on the growing conditions. Understanding these conditions will be useful for readers and other researchers. At the same time, if the authors do not have the necessary information, I do not insist on providing the above data.
Response: Thanks for your great comments and constructive suggestions. We added more detailed information about the time and location of the Lonicerae japonicae flos (JYHX) collection in ‘Section 2.1’. “Fresh Lonicerae japonicae flos (JYHX) were collected on 27 June 2024 in Dakuang Town, Laiyang City, Yantai, Shandong Province” We agree that such details are valuable for reproducibility and metabolomic interpretation. (in lines 84-86)
- Line 86. The authors indicate that they used an internal standard. What substance/substances were used as an internal standard? This information should be provided with the company name.
Response: Thanks for your great comments and constructive suggestions. We added more detailed information about the internal standard in Line 95. “internal standard (2-chlorophenylalanine, 98% purity, J&K Scientific, China)” We would like to express our sincere gratitude once again for your valuable suggestions.
- Results. Paragraph on lines 149-158. When describing the results, do not duplicate the data presented in the figures. Since the relative content of the substances studied by the authors has already been indicated in the diagrams, it is better to describe in the text how many times / by what % the content of one group of substances exceeds the content of another / other groups of substances.
Response: Thanks for your great comments and constructive suggestions. We have revised the Results section (in lines 149-158) to eliminate data duplication with figures, and now present inter-group comparisons using fold-change values as recommended. The revised text is as follows:
“Flavonoids account for the highest proportion, with a content that is 1.4 times that of terpenoids, 1.7–1.8 times that of alkaloids/phenolic acids, and 3.6 times that of lignans/coumarins. As secondary dominant components, terpenoids have an ad-vantage of 1.2–1.3 times over alkaloids and phenolic acids. Flavonoids and terpenoids are the primary secondary metabolites in 'JYHC', accounting for 44.88% of the total together.”
“It can be seen that the main volatile components in ‘JYHC’ are esters and terpenoids, collectively accounting for 37.29% of the total. Their respective concentrations are 1.7 and 1.6 times that of ketones; 1.8 and 1.7 times that of heterocyclic compounds; 2.1 and 2 times that of alcohols; 3 times and 2.9 times that of hydrocarbons, respectively; while trace components such as halogenated hydrocarbons and nitrogen compounds are only 1/26 to 1/138 times that of esters.”
- In the Discussion, mentions to figures and tables given in the text of the manuscript should be added in parentheses at the end of some sentences. For example, '(Fig. 2A)' should be added at the end of the sentence on lines 243-244. Such mentions to figures and tables will help readers understand where in the discussion the results of this study are present, and where the literature data are.
Response: Thanks for your great comments and constructive suggestions. We have carefully revised the discussion section by adding Figure/Table citations in parentheses at relevant positions. We would like to express our sincere gratitude once again for your valuable suggestions.
- Line 340 – 2015 is repeated twice.
Response: Thanks for your great comments and constructive suggestions. We would like to clarify that this citation format follows the journal's specific style guide where the volume number corresponds to the publication year (Volume 2015). According to the journal's citation standard “Year, Volume, page range,” the correct format for this article with Article ID should indeed be “2015, 2015, 905063.”
- Line 376 – not all dots have been placed in the abbreviated title of the journal.
These comments do not reduce the scientific and practical value of the work and are aimed only at improving the quality of the manuscript and the perception of the text by readers.
Response: We apologize for this problem, and the original file was rechecked. We have revised “Crit Rev Food Sci. Nutr.” to “Crit. Rev. Food Sci. Nutr.” in lines 396-397 according to standard academic conventions. We would like to express our sincere gratitude once again for your valuable suggestions. (in lines 423-424)
Round 2
Reviewer 3 Report
Comments and Suggestions for Authors
I am grateful to the authors for revising the manuscript. I have no more comments or suggestions.